# Minigene Splicing Assays Identify 20 Spliceogenic Variants of the Breast/Ovarian Cancer Susceptibility Gene *RAD51C*

**DOI:** 10.3390/cancers14122960

**Published:** 2022-06-15

**Authors:** Lara Sanoguera-Miralles, Elena Bueno-Martínez, Alberto Valenzuela-Palomo, Ada Esteban-Sánchez, Inés Llinares-Burguet, Pedro Pérez-Segura, Alicia García-Álvarez, Miguel de la Hoya, Eladio A. Velasco-Sampedro

**Affiliations:** 1Splicing and Genetic Susceptibility to Cancer, Unidad de Excelencia Instituto de Biología y Genética Molecular, Consejo Superior de Investigaciones Científicas (CSIC-UVa), 47003 Valladolid, Spain; lara.sanoguera@uva.es (L.S.-M.); elena.bueno@uva.es (E.B.-M.); alberto.valenzuela@ibgm.uva.es (A.V.-P.); ines.llinares@alumnos.uva.es (I.L.-B.); aliciaga@ibgm.uva.es (A.G.-Á.); 2Molecular Oncology Laboratory, Hospital Clínico San Carlos, IdISSC (Instituto de Investigación Sanitaria del Hospital Clínico San Carlos), 28040 Madrid, Spain; ada.esteban@salud.madrid.org (A.E.-S.); pedro.perez@salud.madrid.org (P.P.-S.); miguel.hoya@salud.madrid.org (M.d.l.H.)

**Keywords:** hereditary breast and ovarian cancer, cancer susceptibility genes, *RAD51C*, aberrant splicing, functional assay, minigenes, clinical interpretation

## Abstract

**Simple Summary:**

Loss-of-function variants of the *RAD51C* gene are known to confer a risk of breast and ovarian cancers. In this study, we analyzed the impact of *RAD51C* variants on splicing, a highly regulated gene expression step by which introns are removed and exons are sequentially joined. Exon recognition is guided by specific sequences, the 3′ and 5′ splice sites, which define the exon boundaries. Variants of these sequences of susceptibility genes may lead to aberrant splicing and abnormal transcripts that may trigger a disease. Splicing can be tested using a biotechnological tool called minigenes, which mimic the human gene of interest. Thus, we checked 20 *RAD51C* splice-site variants using the minigene mgR51C_ex2-8. We found that they all disrupted the splicing mechanism, and 16 variants could be classified as likely pathogenic. Our findings are clinically actionable, and variant carriers may benefit from tailored prevention protocols and therapies.

**Abstract:**

*RAD51C* loss-of-function variants are associated with an increased risk of breast and ovarian cancers. Likewise, splicing disruptions are a frequent mechanism of gene inactivation. Taking advantage of a previous splicing-reporter minigene with exons 2-8 (mgR51C_ex2-8), we proceeded to check its impact on the splicing of candidate ClinVar variants. A total of 141 *RAD51C* variants at the intron/exon boundaries were analyzed with MaxEntScan. Twenty variants were selected and genetically engineered into the wild-type minigene. All the variants disrupted splicing, and 18 induced major splicing anomalies without any trace or minimal amounts (<2.4%) of the minigene full-length (FL) transcript. Twenty-seven transcripts (including the wild-type and r.904A FL transcripts) were identified by fluorescent fragment electrophoresis; of these, 14 were predicted to truncate the RAD51C protein, 3 kept the reading frame, and 8 minor isoforms (1.1–4.7% of the overall expression) could not be characterized. Finally, we performed a tentative interpretation of the variants according to an ACMG/AMP (American College of Medical Genetics and Genomics/Association for Molecular Pathology)-based classification scheme, classifying 16 variants as likely pathogenic. Minigene assays have been proven as valuable tools for the initial characterization of potential spliceogenic variants. Hence, minigene mgR51C_ex2-8 provided useful splicing data for 40 *RAD51C* variants.

## 1. Introduction

A core set of ten genes significantly increases the lifetime risk of developing breast and/or ovarian cancer (BC/OC), as well as other types of cancer [1]. *RAD51C* (MIM#602774) loss-of-function variants are significantly associated with BC risk (OR = 1.93), while this association is even greater with estrogen-receptor-negative BC, triple-negative BC, and ovarian cancer (OR = 3.99, 5.71, and 5.59, respectively) [2,3,4]. The main isoform of *RAD51C* comprises nine exons and encodes a protein essential for DNA repair by homologous recombination. Biallelic *RAD51C* deleterious variants are also implicated in Fanconi anemia (FANCO) [5,6].

Next-generation sequencing (NGS) technology has allowed great progress in breast/ovarian cancer research and diagnostics but has also increased the number of variants of uncertain clinical significance (VUS), whose role in the disease needs to be clarified. This sort of variant hampers the genetic counseling of patients and decision making in the clinical setting [7]. According to the ClinVar database, around 51% of reported *RAD51C* variants are VUS (https://www.ncbi.nlm.nih.gov/clinvar/?term=RAD51C%5Bgene%5D, (accessed on 21 February 2022)).

The reclassification of VUS is essential to ensure appropriate patient care, and functional assays provide critical information for their interpretation [8,9,10,11]. RNA splicing is one of the gene expression steps that may be impaired by genetic variants [12,13]. This process is controlled by a wide array of motifs, such as the consensus 3′ and 5′ splice sites (3′SS and 5′SS, respectively), the polypyrimidine tract, the branchpoint, and other splicing regulatory elements [14], which represent targets for potential spliceogenic variants. Alterations that result in RNA mis-splicing produce anomalous transcripts and proteins that can trigger a genetic disorder [15,16]. Indeed, a high proportion of VUS of the *BRCA1*, *BRCA2*, *MLH1,* and *MSH2* genes induce splicing disruptions [12,17,18].

In a previous study, we studied 20 *RAD51C* variants from the large-scale sequencing project BRIDGES (http://bridges-research.eu/, accessed on 21 February 2022) in a splicing-reporter minigene that contains exons 2 to 8 [19]. In this study, we bioinformatically analyzed 141 variants reported in the ClinVar database and selected another 20 variants for testing by minigene assays. Finally, we suggested a tentative clinical classification of the BRIDGES variants as per ACMG/AMP (American College of Medical Genetics and Genomics and the Association for Molecular Pathology)-based guidelines.

## 2. Materials and Methods

### 2.1. Ethics Approval

Ethical approval for this study was obtained from the Ethics Committee of the Spanish National Research Council (CSIC, 28/05/2018).

### 2.2. Annotation of DNA and RNA Variants and Transcripts

DNA variants, selected from the ClinVar database (https://www.ncbi.nlm.nih.gov/clinvar/?term=RAD51C%5Bgene%5D (accessed on 21 February 2022)), and alterations at the RNA level were annotated according to the Human Genome Variation Society (HGVS) guidelines (http://varnomen.hgvs.org/, accessed on 21 February 2022) on the basis of the *RAD51C* GenBank sequence NM_058216.3. For clarity, transcripts were also described using a simplified annotation that combines the following symbols [20]: ∆ (skipping of exonic sequences); ▼ (inclusion of intronic sequences); E (exon); p (acceptor shift); and q (donor shift). In addition, the last two symbols (p and q) are always followed by the number of nucleotides inserted or deleted from the 3′SS or 5′SS, respectively. For example, Δ(E2p3) refers to the use of an alternative acceptor site 3 nt downstream of exon 2.

### 2.3. Variant Collection and Filtering

In the ClinVar database, 1316 *RAD51C* variants have been reported (https://www.ncbi.nlm.nih.gov/clinvar/?term=RAD51C%5Bgene%5D (accessed 21 February 2022)). A total of 141 were pre-selected, as they were located at the intronic positions ±10 and the first two and last three nucleotides of *RAD51C* exons 2 to 8 (Appendix A).

Bioinformatics analysis was performed using the Max Ent Scan (MES) algorithm of the R package SpliceSites version 1.0.0 (https://www.bioconductor.org/packages//2.13/bioc/html/spliceSites.html, accessed on 21 February 2022) and SpliceAI version 1.3.1. (https://spliceailookup.broadinstitute.org/, accessed on 21 February 2022) [21,22] (Appendix A). Possible harmful variants were selected according to the following criteria: (i) reduction in MES score by at least 40% (67 variants met this condition); (ii) only one variant per splice-site position unless other events, such as the creation of de novo splice sites or the activation of cryptic ones, were predicted; (iii) prevalence in the ClinVar database, so only variants with at least two records were chosen; and (iv) variants without published reports of splicing assays. Taken together, we finally selected 20 potential spliceogenic variants (totaling 65 ClinVar records) distributed throughout the seven exons of the minigene mgR51C_ex2-8 (Appendix A). SpliceAI (https://spliceailookup.broadinstitute.org/, accessed on 21 February 2022) was also used to predict possible splicing outcomes [22].

### 2.4. Minigene Construction and Mutagenesis

We used a minigene with *RAD51C* exons 2 to 8 (mgR51C_ex2-8) in the splicing vector pSAD [23,24], which was constructed as previously described [19].

The 20 selected variants were incorporated into the wild-type (wt) minigene mgR51C_ex2-8 with the QuikChange Lightning Kit (Agilent, Santa Clara, CA, USA), following the manufacturer’s instructions and using the primers indicated in Appendix A. All constructs were confirmed by sequencing (Macrogen, Madrid, Spain).

### 2.5. Minigene Splicing Assays

MCF-7 and MDA-MB-231 cell culture, transfection, and inhibition of the nonsense-mediated decay were performed as previously described [19,25].

RNA was purified using the Genematrix Universal RNA Purification Kit (EURx, Gdansk, Poland), with on-column DNAse I digestion. Reverse transcription of 400 ng of RNA was performed using the RevertAid First-Strand cDNA Synthesis Kit (Life Technologies, Carlsbad, CA, USA), following the manufacturer’s instructions and employing the vector-specific primer RTPSPL3-RV (5′-TGAGGAGTGAATTGGTCGAA-3′). The resulting cDNA was amplified using Platinum-Taq DNA polymerase (Life Technologies, Carlsbad, CA, USA) and primers SD6-PSPL3_RT-FW (5′-TCACCTGGACAACCTCAAAG-3′) and RTpSAD-RV (CSIC Patent P201231427) (amplicon size: 1062 nt). Samples were subjected to an initial denaturation step at 94 °C for 2 min; followed by 35 cycles of 94 °C/30 s, 60 °C/30 s, and 72 °C/(1 min/kb); and a final extension step at 72 °C for 5 min. RT-PCR products were sequenced by Macrogen (Madrid, Spain), which allowed the characterization of the main variant-induced transcripts. Minor transcripts were annotated according to fluorescent fragment electrophoresis size data (see below).

To quantify the relative amounts of each PCR product, semi-quantitative fluorescent RT-PCRs were carried out in triplicate using a FAM-labeled primer (RTpSAD-RV for minigene cDNA and RTR51C_ex9-RV for cell cDNA) and 26 PCR cycles [26]. Fluorescent products were run with the LIZ-1200 size standard at the Macrogen facility (Seoul, Korea) and analyzed using Peak Scanner software V1.0 (Life Technologies). Only peak heights ≥200 RFU (relative fluorescence units) were considered, and mean peak areas of each transcript and standard deviations were calculated. For clarity, the full protocol is schematized in Appendix A.

### 2.6. ACMG–AMP Clinical Classification of RAD51C Genetic Variants

We classified 20 *RAD51C* genetic variants according to ACMG/AMP-based guidelines [27]. We followed a recently proposed ACMG/AMP point system, a Bayesian framework that outperforms the original classification guidelines and allows for increased flexibility and accuracy in combining different ACMG/AMP criteria and strengths of evidence [28,29]. In this framework, point-based variant classification categories are defined as follows: pathogenic (P) ≥ +10; likely pathogenic (LP) +6 to +9; variant of uncertain significance (VUS) 0 to +5; likely benign (LB) −1 to −6; and benign (B) ≤ −7.

We introduced mgR51C readouts into the classification system as PVS1_O or BP7_O codes of variable evidence strength depending on the splicing outcome (P, supporting (±1 point); M, moderate (±2); strong (±4); very strong (±8)). The use of PVS1_O/BP7_O codes for splicing assays, aimed at highlighting the differences with protein-based functional assays (PS3/BS3 evidence code), was recently introduced by the ClinGen Hereditary Breast, Ovarian, and Pancreatic Cancer Variant Curation Expert Panel (https://www.clinicalgenome.org/docs/clingen-hereditary-breast-ovarian-and-pancreatic-cancer-expert-panel-specifications-to-the-acmg-amp-variant-interpretation/, accessed on 2 May 2022) in its *ATM* rules specifications (HBOPC_ATMv1 specifications).

To deal with complex readouts producing ≥2 transcripts (e.g., a *RAD51C* variant producing two aberrant transcripts, or a leaky variant producing aberrant and full-length transcripts), we developed several ad hoc rules that take into consideration the coding potential of each individual transcript and its relative contribution to the overall expression to reach the appropriate PVS1_O or BP/_O evidence strength. In brief, for each complex readout, we applied the following algorithm: (i) deconvolute mgR51C readouts into individual transcripts; (ii) apply ACMG/AMP evidence classifications to each individual transcript; (iii) produce an overall PVS1_O (or BP7_O) code strength based on the relative contribution of individual transcripts/evidence to the overall expression. Thus, if pathogenic supporting transcripts contribute ≥90% to the overall expression, the PVS1_O_ code is applied (if different transcripts support different pathogenic evidence strengths, the lowest strength contributing >10% to the overall expression is selected for overall evidence strength). Similarly, the BP7_O_ code is applied if benign supporting transcripts contribute ≥90% to the overall expression (if different transcripts support different pathogenic evidence strengths, the lowest strength contributing >10% to the overall expression is selected for overall evidence strength). If neither pathogenic nor benign supporting transcripts contribute ≥90% to the overall expression, the splicing assay is considered to provide no evidence in favor of, or against, pathogenicity. Recently, we used a similar approach to deal with complex *PALB2/ATM* minigene readouts [20,30].

As already justified in previous studies by our group [19,20,25], once experimental splicing data were available, splicing predictive codes PVS1 and PP3 did not contribute to our final classification. Similarly, in HBOPC_ATMv1 specifications, functional splicing codes replace rather than combine with predictive splicing codes.

The rarity code PM2 was considered as per HBOPC_ATMv1 specifications (https://clinicalgenome.org/affiliation/50039/, accessed on 2 May 2022) [30]: (i) allele frequency ≤ 0.01%; (ii) decreasing PM2 evidence strength to ‘supporting’. For allele counting, we interrogated gnomADv2.1 (global). For *RAD51C* variants absent from gnomADv2.1 (no counts), the actual number of interrogated alleles (allele number) was determined using proxy data on the closest available SNP (in all cases, ≤5 nt apart from the variant of interest).

## 3. Results

### 3.1. In Silico Analysis

The ClinVar database contains 1316 variants reported for the *RAD51C* gene, and 141 of them are located at exon/intron boundaries. These variants were bioinformatically analyzed with MES according to the standards indicated in the Materials and Methods section (Appendix A). Twenty variants from exons 2 to 8 were selected for functional assays (Table 1, Appendix A, Figure 1a). Fourteen variants affected intronic ±1,2 positions (c.146-4_146-2del, c.404+2T>C, c.405-1G>C, c.571+1del, c.572-1G>C, c.705+1G>A, c.706-1G>T, c.837+1G>T, c.838-2A>G, c.904+1G>T, c.905-3_906del, c.905-2del, c.965+1G>A, and c.966-1G>C); another four altered ±3 nucleotides (c.146-3C>G, c.404+3A>G, c.572-3C>G, and c.705+3A>G); one substituted the last exon nucleotide (c.904G>A); and another affected the +4 intronic position (c.837+4_837+7del).

Ten of these selected variants were predicted to impair the acceptor site and another ten were expected to impact the donor site. Seven variants (c.146-4_146-2del, c.405-1G>C, c.571+1del, c.705+3A>G, c.706-1G>T, c.905-3_906del, and c.966-1G>C) were predicted to impair the SS and simultaneously create new SSs or strengthen nearby cryptic ones, according to MES. In addition, according to spliceAI, four variants were predicted to promote the use of cryptic splice sites (c.404+2T>C, c.404+3A>G, c.904G>A, and c.904+1G>T).

### 3.2. Functional STUDY

The RNA analysis of the wt minigene revealed the full-length (mgFL) transcript (V1-*RAD51C* exons 2 to 8-V2) and traces (1.4%) of an uncharacterized transcript of 1106 nt (Table 1, Figure 1b), as previously reported [19].

All variants altered splicing: 18 produced no traces of the mgFL transcript or almost undetectable levels (<2.4%, c.904G>A), and it was detected in low proportions (26.3% and 21.3%) in the 2 remaining alterations (c.404+3A>G and c.705+3A>G, respectively) (Table 1). Finally, to check the minigene reproducibility in other cell lines, 3 out of the 20 selected variants (c.405-1G>C, c.706-1G>T, and c.904G>A), as well as the wt minigene, were also tested in the triple-negative breast cancer MDA-MB-231 cells. Fluorescent fragment electrophoresis revealed that MDA-MB-231 cells mimicked the splicing profiles found in MCF-7 cells (Figure 2).

### 3.3. Transcript Analysis and ACMG/AMP-Based Interpretation

Semi-quantitative fluorescent RT-PCR revealed 27 different aberrant splicing events, including 2 minigene FL transcripts (wt and c.904G>A) (Figure 3, Appendix A). Nineteen of them could be characterized, and the remaining eight uncharacterized transcripts appeared in low proportions (≤4.7%) and represented, at most, 6.2% of the overall minigene expression (Table 1). A high-resolution image of the fluorescent fragment electrophoresis is illustrated in Figure 1c, where transcripts with small size differences (i.e., 1, 3 nt) can be distinguished. Alternative site usage was the most frequent splicing event; specifically, four aberrant transcripts used cryptic 3′SS (Δ(E2p3), Δ(E3p7), Δ(E5p10), and ▼(E8p3)), and six used alternative 5′SS (▼(E2q27)-a, ▼(E2q27)-b, Δ(E2q175), Δ(E3q1), ▼(E6q4)-a, and ▼(E6q4)-b). The annotations ▼(E2q27)-a and -b reflect the same splicing event (donor shift of exon 2, r.404+27) with different sequences (r.404+2C and r.404+3G, respectively). Similarly, ▼(E6q4)-a and -b indicate the donor shift of exon 6 (r.904+4) with different sequences (r.904A and r.904+1U, respectively). The remaining seven events represented exon skipping (Δ(E2), Δ(E3), Δ(E4), Δ(E5), Δ(E6), Δ(E7), and Δ(E8)).

Of the 19 characterized transcripts, 14 introduced premature termination codons (PTC; PTC transcripts), and of these, 10 were predicted to be degraded by the nonsense-mediated decay pathway (NMD; PTC-NMD transcripts), which is considered convincing evidence of deleteriousness (Appendix A). Following the ACMG/AMP’s proposed PVS1 decision-tree rationale [31], all PTC-NMD transcripts (Table 1) were classified as very strong evidence of pathogenicity (Table 2). The four PTC non-NMD transcripts, ▼(E6q4)-a (p.Gly302SerFs*47), ▼(E6q4)-b (p.Gly302ValFs*47), Δ(E7) (p.Glu303TrpFs*41), and Δ(E8) (p.Arg322SerFs*22), target RAD51C regions critical for protein function. According to the PVS1 decision-tree rationale, these four PTC transcripts should be considered as strong evidence of pathogenicity. However, these alterations remove β strands 6 to 9 (7 to 9 in the case of Δ(E8)) and the nuclear localization signal [32,33]. The integrity of the β sheet is important for maintaining the overall fold of the RAD51C protein and the interaction with RAD51B, so alterations to any β strand of RAD51C should be considered deleterious [33]. Further, structural features (the order of the β strands in space is not the same as their order in sequence) predict that proteins lacking any single β strand would fail to form the β sheet, resulting in the collapse of the protein core and the misfolding of the protein [33]. Moreover, the missense variant p.Arg312Trp (β strand 6) has been shown to impair RAD51C function [34]. Considering these data altogether, we decided to upgrade the pathogenic evidence strength from strong to very strong (Table 1 and Table 2). In keeping with this, various PTC variants are classified as pathogenic/likely pathogenic by multiple submitters (no conflicts) in ClinVar.

Three aberrant transcripts kept the open reading frame (Δ(E2p3), Δ(E5), and ▼(E8p3)) (Table 1 and Table 2). Δ(E2p3) is a physiological alternative isoform [35] that deletes the conserved amino acid Glu49 (Appendix A). Lacking any evidence other than a deleterious PROVEAN score (- 10.29), we determined that this in-frame transcript provides pathogenic evidence with supporting strength (as per PP3). The predicted protein product of Δ(E5) (p.Arg237_Val280del) deletes the Walker-B domain (β strand 4) and β strand 5. In addition, 26 out of the 44 amino acids encoded by this exon are conserved in vertebrates (Appendix A) [19]. Finally, the exon 5 missense variant c.773G>A (p.Arg258His) is classified as likely pathogenic in ClinVar, because it was found as a biallelic mutation in multiple Fanconi anemia patients of a single family [6]. Altogether, these observations suggest that Δ(E5) is a loss-of-function transcript that should be catalogued as very strong evidence of pathogenicity (P_VS, +8 points, Table 2). Finally, ▼(E8p3) removes the conserved amino acid Arg322 (β strand 7) and inserts Ser and Thr (p.Arg322delinsSerThr) (Appendix A). Based on a deleterious PROVEAN score (−11.94), we determined that this in-frame transcript provides pathogenic evidence with supporting strength (as per PP3).

Finally, one mgFL-transcript carried the missense variant c.904G>A/p.Gly302Arg, where Gly302 is conserved in vertebrates but does not affect a known protein functional domain. In addition, the metapredictor REVEL does not support the pathogenicity of this missense variant (0.5) [36]. Another nucleotide substitution (c.904G>C), resulting in the same missense variant (p.Gly302Arg), is considered a VUS in ClinVar (REVEL = 0.5).

Thus, mgFL-transcript-c.904G>A does not support any evidence of pathogenicity (P_N/A).

We classified all 20 *RAD51C* variants according to ACMG-AMP-based classification guidelines, integrating mgR51C data as PVS1_O/BP7_O evidence codes (as indicated above) and the rarity code PM2 (as indicated in Materials and Methods, Table 2). The PM3 evidence (in trans with a pathogenic variant in a recessive disorder) did not contribute to the final classification. Unsurprisingly (FANCO is an extremely rare FA complementation group) [37], none of the tested variants have been identified in Fanconi anemia patients (ClinVar and Global Variome share LOVD databases and literature searches). Similarly, the BS2 evidence (in trans with a pathogenic variant in a healthy individual) did not contribute to the final classification of our tested variants. Finally, we decided that some pathogenic (PS2, PM1, PM6, PP2, PP4) and benign (BP1, BP3, BP5) codes were not applicable to the classification of *RAD51C* variants.

Thus, 16 variants were classified as likely pathogenic and 4 as VUS (Table 2). Compared with the ClinVar classification, four variants (c.146-3C>G, c.572-3C>G, c.837+4_837+7del, and c.904G>A) were upgraded from VUS to likely pathogenic, while two variants (c.146-4_146-2del and c.966-1G>C) were downgraded from likely pathogenic to VUS (Table 2).

## 4. Discussion

About 40% of all variants reported in the ClinVar database are variants of uncertain significance (https://www.ncbi.nlm.nih.gov/clinvar?term=%22clinvar_all%22[Filter], accessed on 21 February 2022). Variants of uncertain significance pose a challenge for genetic counseling testing as they are considered a negative result, and so the risk assessment of VUS-carrier patients is exclusively based on family history [8,38]. A significant fraction of VUS impair pre-mRNA splicing, which makes transcript analysis a mandatory step for determining their pathogenicity [39].

Conversely, many splicing variants are classified as likely pathogenic or pathogenic because they target the canonical ±1, 2 splice-site positions. While it is true that most of these variants will impact splicing, the resulting alteration is not necessarily pathogenic. For example, variants affecting the 3′SS of *RAD51C* exon 8, such as c.966-2A>G (functionally analyzed in our previous *RAD51C* study [19]) and c.966-1G>C (studied here), are classified as likely pathogenic by ClinVar because they alter the -2 and -1 positions, respectively. Nevertheless, these variants cause a 3 nt insertion (one amino acid) with an unknown impact on protein function, and so they are classified as VUS (Table 2). This observation underlines the importance of the functional testing of suspicious splicing variants [14,40].

Here, we focused on potentially spliceogenic *RAD51C* variants reported in the ClinVar database. Functional studies were performed using a hybrid minigene (mgR51C_ex2-8) that has proven to be a powerful and reliable tool for testing variant-splicing outcomes in the absence of patient RNA [19]. The major advantages of minigene-based assays are: (a) no need for patient samples; (b) no interference from the wt allele, as occurs in patient RNA, so all the observed transcripts are generated by the variant; (c) assays can be performed on disease-relevant cell types; and (d) a single construct allows the study of multiple variants. Indeed, this *RAD51C* minigene has allowed the functional analysis of a total of 40 variants to date (Table 3), but the functional analysis of any candidate variant located at exons 2 to 8 would be possible. Finally, the high sensitivity and resolution of the fluorescent fragment electrophoresis, which facilitated the detection of rare transcripts and resolved small size differences between them, is also worth mentioning.

All variants tested impair splicing, underlining the specificity of our selection criteria. MES accurately predicted all the splicing disruptions, indicating that the selection criterion of a ≤40% MES score reduction was appropriate. In general, splicing outcomes predicted by SpliceAI were precise, except for variants c.705+3A>G, c.904+1G>T, and c.966-1G>C (Appendix A; Table 1). Indeed, it is realistically impossible to forecast all variant-induced transcripts, given that most variants generate ≥2 different transcripts, and so, as indicated above, RNA assays are compulsory to clinically classify them. Fluorescent fragment electrophoresis allowed us to detect 27 different RNA isoforms, 19 of which could be characterized (Table 1, Figure 1 and Figure 3, Appendix A). Seven of them were generated by exon skipping (Δ(E2), Δ(E3), Δ(E4), Δ(E5), Δ(E6), Δ(E7), and Δ(E8)); four by alternative 3′SS usage (Δ(E2p3), Δ(E3p7), Δ(E5p10), and ▼(E8p3)); and six involved the use of cryptic/de novo 5′SS (▼(E2q27)-a,b, Δ(E2q175), Δ(E3q1), and ▼(E6q4)-a,b). Remarkably, 10 variant-induced transcripts (Δ(E2p3), ▼(E2q27), Δ(E2q175), Δ(E2), Δ(E3), Δ(E4), Δ(E5), Δ(E7), Δ(E8), and ▼(E8p3)) had been previously characterized as naturally occurring isoforms of *RAD51C* [35], suggesting that alternative events may match, at least in part, variant-splicing profiles.

Only two variants (c.404+3A>G and c.705+3A>G) displayed 26% and 21% of the mgFL transcript, respectively. Unfortunately, the minimal amount of *RAD51C* expression required to confer tumor-suppressor haplosufficiency is unknown, and so these splicing assays were not considered informative (PVS1_O_N/A). Interestingly, both alter the intronic position +3 (Table 1). Remarkably, +3A is the main nucleotide at this position (59.1%), but +3G is also relatively frequent (34.2%), highlighting the importance of both nucleotides for 5′-SS selection [41]. Our previous studies of +3 changes to G revealed diverse effects. While some exhibited no trace of the mgFL transcript (or small amounts ≤10%) (*BRCA2* c.67+3A>G, c.631+3A>G, and c.8487+3A>G; *BRCA1* c.212+3A>G; and *PALB2* c.3113+3A>G), others did not even affect the splicing process (or showed weak effects) (*BRCA2* c.7435+3A>G and *PALB2* c.2834+3A>G) [12,42,43,44]. Previous papers have suggested that such transitions are deleterious when the affected 5′SS have non-consensus nucleotides in the adjacent +4 and +5 positions [45,46]. However, we found that most +3A>G variants do not follow this simple +4,+5 mismatch rule [20,43,44].

We also focused our attention on substitutions of +2T>C, by means of which a canonical GT 5′ SS is converted into an atypical GC 5′ SS that accounts for less than 1% of human donor sites [47]. It has been reported that about 15–18% of +2T>C changes retain the activity of the donor site [48], inducing between 1 and 84% of full-length transcripts. Remarkably, neither of the two genetic alterations that introduce a cytosine at position +2 (c.404+2T>C and c.837+2T>C) use the de novo atypical GC dinucleotide (Table 2), e.g., the *PALB2* variant c.48+2T>C [19,20]. Conversely, we previously showed that *PALB2* c.108+2T>C generated an active GC-5′SS that produced 85% of full-length transcripts [20]. This feature may be related to the high sequence conservation of the other splice-site positions (CAG|GCAAGT).

On the other hand, c.966-1G>C mainly induced the use of an alternative 3′SS (▼(E8p3), 79%), which we had also detected in variants c.966-3C>A, c.966-2A>G, and c.966-2A>T, though in lower amounts (6–11%) [19]. As indicated above, the 3 nt insertion ▼(E8p3) represents four different transcripts and four different protein products (Arg deletion/SerThr insertion, Arg duplication, Arg deletion/SerGly insertion, and Arg deletion/SerTrp insertion), which hinder transcript interpretation to an even greater extent.

## 5. Conclusions

We tested a total of 40 *RAD51C* variants in the minigene mgR51C_ex2-8, of which 39 impaired splicing and 36 were associated with severe splicing aberrations (Table 3). Thirty-one variants were classified as likely pathogenic/pathogenic as per ACMG/AMP-based guidelines, while nine were catalogued as VUS. Moreover, according to ClinVar records of 34 reported variants (including those of our previous study) [19], the mgR51C readouts changed the clinical interpretation of 12 variants: 9 VUS were upgraded to likely pathogenic and 3 LP variants were downgraded to VUS. Both changes are critical for genetic counseling and decision making in the clinical setting, reaffirming the value of minigene assays. Finally, it is critical to define the minimal amount of *RAD51C* required to maintain gene function. Hence, it is conceivable that the variants with the vast majority of inactivating transcripts, such as c.966-3C>A, c.966-2A>G, and c.966-2A>T (>86% of PTC transcripts), might be reclassified as likely pathogenic or pathogenic.

## Figures and Tables

**Figure 1 cancers-14-02960-f001:**
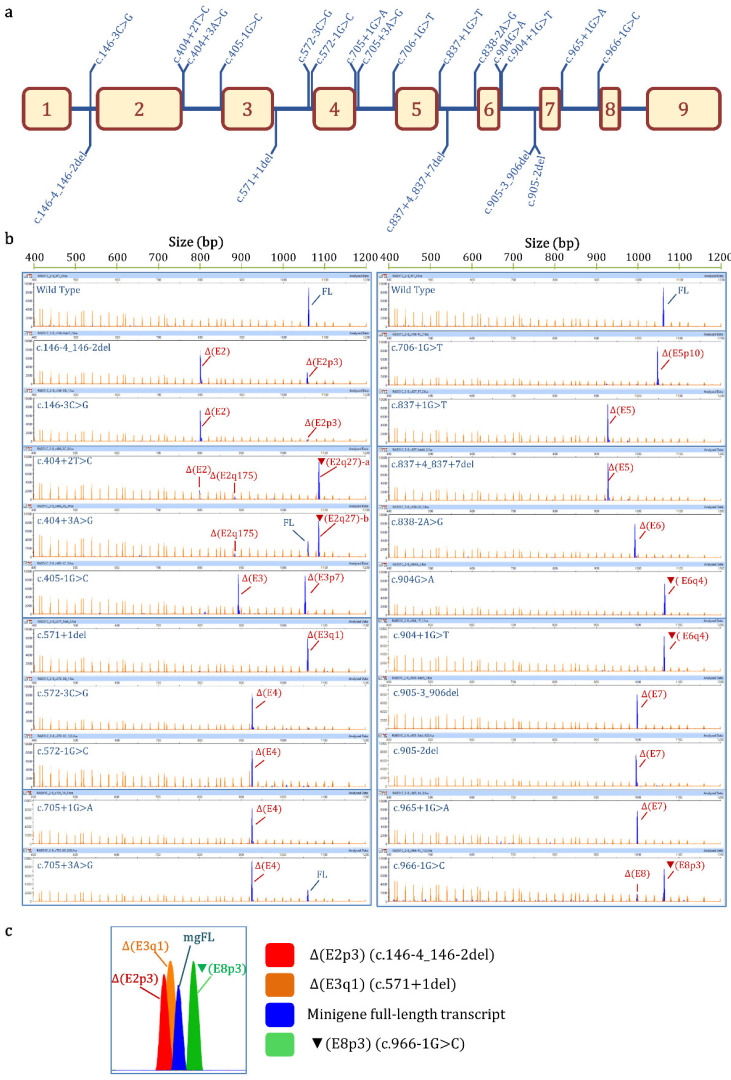
Minigene splicing assays of selected ClinVar variants. (**a**) Map of variants in the minigene mgR51C_ex2-8. (**b**) Fluorescent fragment analysis of the ClinVar variants. FAM-labelled products (blue peaks) were run with LIZ1200 (orange peaks) as size standard (FL, minigene full-length transcript). (**c**) High-resolution image of fluorescent fragment electrophoresis that discriminates minimal size differences between transcripts.

**Figure 2 cancers-14-02960-f002:**
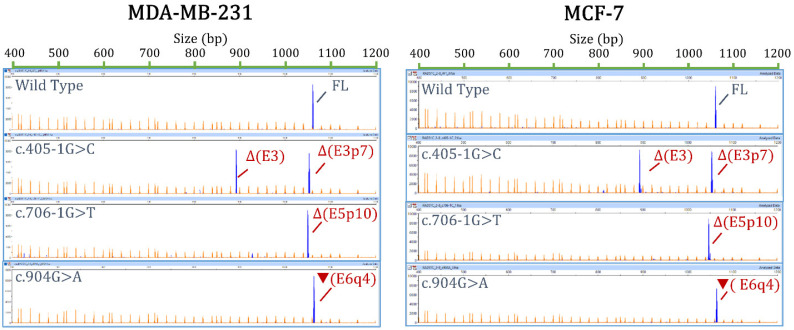
Reproducibility of splicing assays in MDA-MB-231 (**left**) and MCF-7 (**right**) cells. The wild-type and mutant minigenes of c.405-1G>C, c.706-1G>T, and c.904G>A were tested in MCF-7 and MDA-MB-231 cells. RT-PCR products were run on agarose gels.

**Figure 3 cancers-14-02960-f003:**
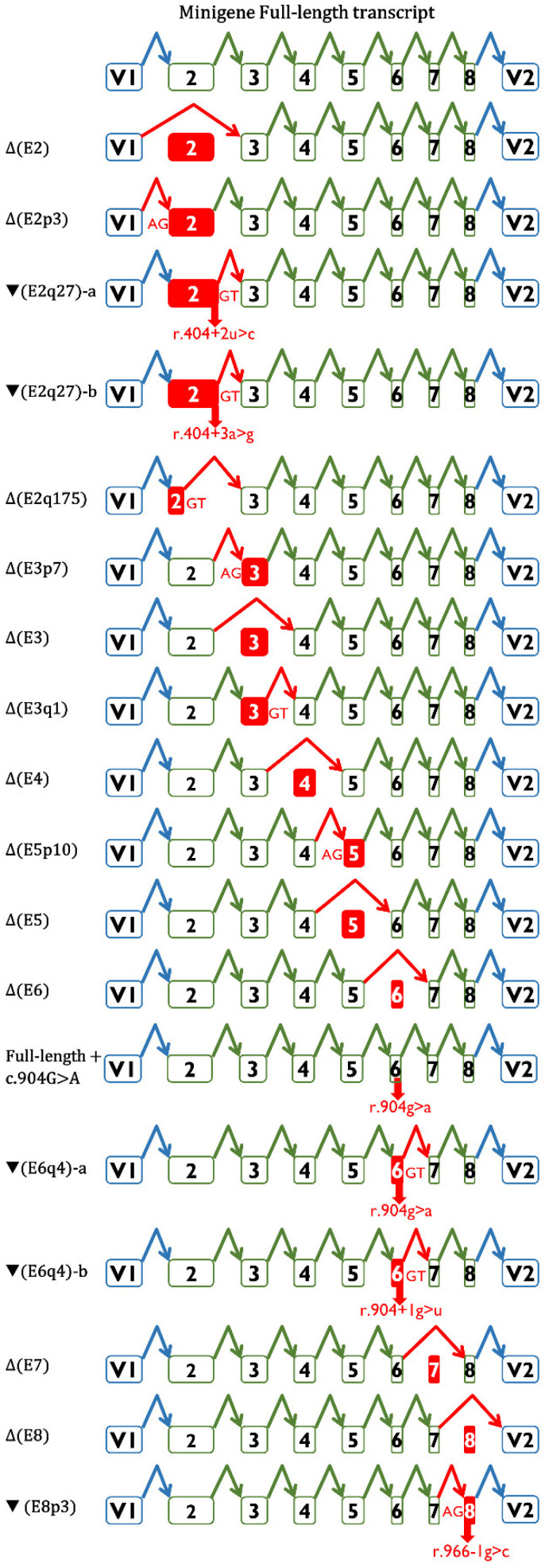
Splicing isoforms generated by *RAD51C* variants. Diagrams of the splicing reactions. Exons and the splicing reactions are indicated by boxes and elbow arrows, respectively. Anomalous events and exons are indicated in red.

**Table 1 cancers-14-02960-t001:** Bioinformatics analysis and splicing outcomes of *RAD51C* canonical splice variants.

Variant (HGVS) ^1^	Bioinformatics Summary ^2^	Transcripts ^3^			
		Canonical	PTC	In-Frame	Uncharacterized
Wild-type		98.6% ± 0.2%			1106 nt (1.4% ± 0.2%)
c.146-4_146-2del	[−]3′SS (9.5→ −0.8) [+]3′SS (5.9) 3 nt downstream	-	Δ(E2): 73.8% ± 0.8%	Δ(E2p3): 25.1% ± 0.4%	657nt (1.1% ± 0.9%)
c.146-3C>G	[−]3′SS (9.5→ 1.9)	-	Δ(E2): 94.8% ± 0.9%	Δ(E2p3): 5.2% ± 0.9%	
c.404+2T>C	[−]5′SS (4.8→ −3.0) Cr. 5′SS (5.4) 27nt downstream	-	▼(E2q27)-a: 77.2% ± 1.3% Δ(E2): 16.7% ± 0.3% (E2q175): 4.7% ± 0.1%		657 nt (1.4% ± 1.2%)
c.404+3A>G	[−]5′SS (4.8→ 0.6) Cr. 5′SS (5.4) 27 nt downstream	26.3% ± 0.4%	▼(E2q27)-b: 66.4% ± 1.6% Δ(E2q175): 5.4% ± 0.3%		657 nt (1.9% ± 1.6%)
c.405-1G>C	[−]3′SS (7.7→ −0.4) [+]3′SS (4.2) 7 nt downstream	-	Δ(E3p7): 48.9 ± 1.6% Δ(E3): 48.2% ± 1.2%		813 nt (2.9% ± 0.3%)
c.571+1del	[−]5′SS (10.5→ −14.1) [+]5′SS (11.1) 1 nt upstream	-	Δ(E3q1): 98.4% ± 1.4%		800 nt (1.6% ± 1.4%)
c.572-3C>G	[−]3′SS (7.4→ −1.4)	-	Δ(E4): 94.7% ± 0.3%		1063 nt (3.0% ± 0.2%) 1008 nt (2.3% ± 0.1%)
c.572-1G>C	[−]3′SS (7.4→ −0.6)	-	Δ(E4): 93.8% ± 0.0%		1008 nt (3.2% ± 0.0%) 1063 nt (1.5% ± 0.0%) 976 nt (1.5% ± 0.0%)
c.705+1G>A	[−]5′SS (9.1→ 0.9)	-	Δ(E4): 100%		
c.705+3A>G	[↓]5′SS (9.1→ 4.6) [+]5′SS (6.1) 2 nt downstream	21.3% ± 1.3%	Δ(E4): 78.7% ± 1.3%		
c.706-1G>T	[−]3′SS (11.1→ 2.5) [+]3′SS (4.3) 10 nt downstream	-	Δ(E5p10): 100%		
c.837+1G>T	[−]5′SS (8.6→ 0.1)	-		Δ(E5): 95.3% ± 0.4%	976 nt (4.7% ± 0.4)
c.837+4_837+7del	[−]5′SS (8.6→ −8.9)	-		Δ(E5): 100%	
c.838-2A>G	[−]3′SS (10.2→ 2.2)	-	Δ(E6): 98.4% ± 1.4%		590 nt (1.6% ± 0.4)
c.904G>A (p.Gly302Arg)	[−]5′SS (5.6→ 1) Cr. 5′SS (6.2) 4 nt downstream	2.4% ± 0.1%	▼(E6q4)-a: 97.6% ± 0.1%		
c.904+1G>T	[−]5′SS (5.6→ −3.0) Cr. 5′SS (6.2) 4 nt downstream	-	▼(E6q4)-b: 100%		
c.905-3_906del	[−]3′SS (8.2→ −8.6) [+]3′SS (4.5) 7 nt downstream	-	Δ(E7): 100%		
c.905-2del	[−]3′SS (8.2→ 2.1)	-	Δ(E7): 100%		
c.965+1G>A	[−]5′SS (8.7→ 0.5)	-	Δ(E7): 100%		
c.966-1G>C	[−]3′SS (7.3→ −0.8) [↑] Cr. 3′SS (6.8) 3 nt upstream	-	Δ(E8): 20.6% ± 0.1%	▼(E8p3): 79.4% ± 0.1%	

^1^ Variants without any trace (or ≤5%) of the full-length transcript are underlined. ^2^ [−] Site disruption; [+] new site; [↑] the strength of the SS is increased; Cr. cryptic splice sites of interest; ^3^ PTC, premature termination codon. The transcripts are named as follows: ∆ (skipping of exonic sequences); ▼ (inclusion of intronic sequences); E (exon); and, when necessary, p (acceptor shift) and q (donor shift) + nt inserted or deleted.

**Table 2 cancers-14-02960-t002:** Proposed clinical classification of *RAD51C* variants according to ACMG/AMP-based criteria.

HGVS ^1^	ClinVar Accession	PVS1 ^2^	PP3/BP4 ^3^	PVS1_O/BP7_O mgR51C_ex2-8 ^4^	PM2 ^5^	pSAD-based ACMG/AMP-like Classification ^6^	ClinVar Classification ^7^
c.146-4_146-2del	VCV000482181.5	PVS1	N/A	PVS1_O_P (+1) [Δ(E2): 74%, P_VS+ Δ(E2p3): 26%, P_P]	(0/250,394) PM2_P (+1)	VUS (+2)	LP
c.146-3C>G	VCV000484752.4	N/A	(−79.4%) PP3	PVS1_O_VS (+8) [Δ(E2): 95% P_VS + Δ(E2p3): 5% P_P]	(0/250,394) PM2_P (+1)	LP (+9)	VUS
c.404+2T>C	VCV000182835.10	PVS1	N/A	PVS1_O_VS (+8) [▼(E2q27)-a: 77%, P_VS Δ(E2): 17%, P_VS (E2q175): 5%, P_VS]	(1/246,102) PM2_P (+1)	LP (+9)	P/LP
c.404+3A>G	VCV000409857.4	N/A	(−92.5%) PP3	PVS1_O_N/A BP7_O-N/A [▼(E2q27): 66%, P_VS + Δ(E2q175): 5%, P_VS + FL, 27% B_S]	(0/246,102) PM2_P (+1)	VUS (+1)	VUS
c.405-1G>C	VCV000141823.5	PVS1	N/A	PVS1_O_VS (+8) [Δ(E3p7): 49%, P_VS Δ(E3): 48%, P_VS]	(0/251,476) PM2_P (+1)	LP (+9)	LP
c.571+1del	VCV000482176.8	PVS1	N/A	PVS1_O_VS (+8) [Δ(E3q1): 98%, P_VS]	(1/251,452) PM2_P (+1)	LP (+9)	LP
c.572-3C>G	VCV000633386.5	N/A	(−99.4%) PP3	PVS1_O_VS (+8) [Δ(E4): 95%, P_VS]	(0/251,198) PM2_P (+1)	LP (+9)	VUS
c.572-1G>C	VCV000480497.10	PVS1	N/A	PVS1_O_VS (+8) [Δ(E4): 94%, P_VS]	(0/251,220) PM2_P (+1)	LP (+9)	P/LP
c.705+1G>A	VCV000230577.9	PVS1	N/A	PVS1_O_VS (+8) [Δ(E4): 100%, P_VS]	(0/251,038) PM2_P (+1)	LP (+9)	LP
c.705+3A>G	VCV000241775.7	N/A	(−31.0%) PP3	PVS1_O_N/A BP7_O_N/A [Δ(E4): 79%, P_VS + FL: 21%, B_S]	(0/250,946) PM2_P (+1)	VUS (+1)	VUS
c.706-1G>T	VCV000452310.4	PVS1	N/A	PVS1_O_VS (+8) [Δ(E5p10): 100%, P_VS]	(0/282,746) PM2_P (+1)	LP (+9)	LP
c.837+1G>T	VCV000241779.3	PVS1	N/A	PVS1_O_VS (+8) [Δ(E5): 95%, P_VS]	(0/251,374) PM2_P (+1)	LP (+9)	LP
c.837+4_837+7del	VCV000128212.8	N/A	(−100.0%) PP3	PVS1_O_VS (+8) [Δ(E5): 100%, P_VS]	(0/251,374) PM2_P (+1)	LP (+9)	LP(1);VUS(2)
c.838-2A>G	VCV000480508.3	PVS1	N/A	PVS1_O_VS (+8) [Δ(E6): 98%, P_VS]	(0/250,982) PM2_P (+1)	LP (+9)	LP
c.904G>A	VCV000478781.9	N/A	(−89.3%) PP3	PVS1_O_VS (+8) [▼(E6q4)-a: 98%, P_VS]	(0/250,832) PM2_P (+1)	LP (+9)	LP(1);VUS(3)
c.904+1G>T	VCV000480510.7	PVS1	N/A	PVS1_O_VS (+8) [▼(E6q4)-b: 100%, P_VS]	(0/250,832) PM2_P (+1)	LP (+9)	LP
c.905-3_906del	VCV000182846.7	PVS1	N/A	PVS1_O_VS (+8) [Δ(E7): 100%, P_VS]	(0/282,730) PM2_P (+1)	LP (+9)	P/LP
c.905-2del	VCV000230587.7	PVS1	N/A	PVS1_O_VS (+8) [Δ(E7): 100%, P_VS]	(0/282,730) PM2_P (+1)	LP (+9)	LP
c.965+1G>A	VCV000182838.5	PVS1	N/A	PVS1_O_VS (+8) [Δ(E7): 100%, P_VS]	(0/ 251118) PM2_P (+1)	LP (+9)	LP
c.966-1G>C	VCV000851327.3	PVS1	N/A	PVS1_O_P (+1) [Δ(E8): 21%, P_VS + ▼(E8p3): 79%, P_P]	(0/251,358) PM2_P (+1)	VUS (+2)	LP

^1^ NM_058216.3. ^2^ PVS1 (pathogenic very strong). ^3^ PP3/BP4 (computational evidence supports a deleterious effect/suggests no impact). ^4^ PVS1_O code strength derived from mgR51C readouts (P, supporting (±1 point); M, moderate (±2); strong (±4); very Strong (±8)). Percentages of transcripts from Table 1 were rounded. ^5^ Rarity evidence PM2 downgraded to supporting strength, as per ClinGen ATM expert panel ACMG-AMP specifications. For rarity evidence, we used the global gnomADv2.1 data. ^6^ Predictive evidence codes (PVS1/PP3/BP4) are excluded from our pSAD-based ACMG/AMP-like classification approach. Pathogenic (P) ≥ +10; likely pathogenic (LP) +6 to +9; variant of uncertain significance (VUS) 0 to +5; likely benign (LB) −1 to −6; and benign (B) ≤ −7. ^7^ ClinVar as of 16 February 2022. For conflicting interpretations of pathogenicity, the number of submitters supporting each interpretation is indicated.

**Table 3 cancers-14-02960-t003:** Summary of the 40 variants tested in minigene mgR51C_ex2-8.

*RAD51C* Variant	Splicing Motif ^1^	Splicing Outcome ^2^	Clinical Interpretation
Exon 2			
c.146-4_146-2del	[±]3′SS	Δ(E2): 73.8%; Δ(E2p3): 25.1%	VUS
c.146-3C>T	[−]3′SS	100% mgFL-transcript	VUS ^3^
c.146-3C>G	[−]3′SS	Δ(E2): 94.8%; Δ(E2p3): 5.2%	Likely Pathogenic
c.404G>A	[−]5′SS	▼(E2q27): 69.3%; Δ(E2q175): 19.9%; Δ(E2q22): 4.3%; Δ(E2): 2.4%	Likely Pathogenic ^3^
c.404+2T>C	[−]5′SS	▼(E2q27): 77.2%; Δ(E2): 16.7%; (E2q175): 4.7%	Likely Pathogenic
c.404+3A>G	[−]5′SS	▼(E2q27): 66.4%; mgFL: 26.3%; Δ(E2q175): 5.4%	VUS
Exon 3			
c.405-6T>A	[±]3′SS/Pyr	▼(E3p4):95.2%; Δ(E3): 4.8%	Likely Pathogenic ^3^
c.405-1G>C	[±]3′SS	Δ(E3p7): 48.9%; Δ(E3): 48.2%	Likely Pathogenic
c.571+1del	[±]5′SS	Δ(E3q1): 98.4%	Likely Pathogenic
c.571+4A>G	[±]5′SS	Δ(E3): 76.5%; ▼(E3q4): 11.6%; FL: 5.4%; Δ(E3q114): 4.0%	Likely Pathogenic ^3^
c.571+5G>A	[−]5′SS	Δ(E3): 91.5%; Δ(E3q114): 4.8%	Pathogenic ^3^
Exon 4			
c.572-3C>G	[−]3′SS	Δ(E4): 94.7%	Likely Pathogenic
c.572-1G>C	[−]3′SS	Δ(E4): 93.8%	Likely Pathogenic
c.572-1G>T	[−]3′SS	Δ(E4): 93.4%	Likely Pathogenic ^3^
c.705G>T	[−]5′SS	Δ(E4): 100%	Likely Pathogenic ^3^
c.705+1G>A	[−]5′SS	Δ(E4): 100%	Likely Pathogenic
c.705+3A>G	[−]5′SS	Δ(E4): 78.7%; mgFL: 21.3%	VUS
c.705+5G>C	[−]5′SS	mgFL: 51.6%; Δ(E4): 48.4%	VUS ^3^
Exon 5			
c.706-2A>C	[±]3′SS	Δ(E5p10): 91.4%; Δ(E5): 4.0%; Δ(E5p52): 1.8%	Likely Pathogenic ^3^
c.706-2A>G	[±]3′SS	Δ(E5): 65.4%; Δ(E5p10): 33.5%	Pathogenic ^3^
c.706-1G>T	[±]3′SS	Δ(E5p10): 100%	Likely Pathogenic
c.837+1G>T	[−]5′SS	Δ(E5): 95.3%	Likely Pathogenic
c.837+2T>C	[−]5′SS	Δ(E5): 89.3%; Δ(E4_5): 2.2%	Likely Pathogenic ^3^
c.837+4_837+7del	[−]5′SS	Δ(E5): 100%	Likely Pathogenic
Exon 6			
c.838-2A>G	[−]3′SS	Δ(E6): 98.4%	Likely Pathogenic
c.904G>A	[−]5′SS	▼(E6q4): 97.6%; FL: 2.4%	Likely Pathogenic
c.904+1G>T	[−]5′SS	▼(E6q4): 100%	Likely Pathogenic
Exon 7			
c.905-3_906del	[−]3′SS	Δ(E7): 100%	Likely Pathogenic
c.905-3C>G	[−]3′SS	Δ(E7): 98.1%; Δ(E7_8): 1.9%	Likely Pathogenic ^3^
c.905-2del	[−]3′SS	Δ(E7): 100%	Likely Pathogenic
c.905-2A>C	[−]3′SS	Δ(E7): 97.4 %	Pathogenic ^3^
c.905-2_905-1del	[−]3′SS	Δ(E7): 100%	Pathogenic ^3^
c.965+1G>A	[−]5′SS	Δ(E7): 100%	Likely Pathogenic
c.965+5G>A	[−]5′SS	Δ(E7): 100%	Likely Pathogenic ^3^
Exon 8			
c.966-3C>A	[−]3′SS	Δ(E8): 86.8%; ▼(E8p3): 9.7%; FL: 2%	VUS ^3^
c.966-2A>G	[±]3′SS	Δ(E8): 86.7%; ▼(E8p3):11.0%	VUS ^3^
c.966-2A>T	[±]3′SS	Δ(E8): 89.1%; ▼(E8p3):5.9%	VUS ^3^
c.966-1G>C	[±]3′SS	▼(E8p3): 79.4%; Δ(E8): 20.6%	VUS
c.1026+5_1026+7del	[−]5′SS	Δ(E8): 79.5%; Δ(E8q18):13.8%; ▼(E8q41): 3.3%	Pathogenic ^3^
c.1026+5G>T	[−]5′SS	Δ(E8): 78.0%; Δ(E8q18):18.7%; ▼(E8q44): 1.4%	Likely Pathogenic ^3^

^1^ [−] Site disruption; [+] new site; [±] simultaneous creation/strengthening of cryptic site and disruption; Pyr, polypyrimidine tract. ^2^ Only characterized transcripts are shown in this column; transcripts are named as follows: ∆ (skipping of exonic sequences); ▼ (inclusion of intronic sequences); E (exon); and, when necessary, p (acceptor shift) and q (donor shift) + nt inserted or deleted; FL, full-length. ^3^ ACMG/AMP-based interpretation according to Sanoguera-Miralles et al., 2020 [19].

## Data Availability

All sequencing and fragment analysis data are available at http://hdl.handle.net/10261/270934; https://doi.org/10.20350/digitalCSIC/14662, accessed on 15 June 2022.

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
