# Peer review of "Minigene Splicing Assays Identify 20 Spliceogenic Variants of the Breast/Ovarian Cancer Susceptibility Gene RAD51C"

_cancers, 2022, doi:10.3390/cancers14122960_

Round 1
Reviewer 1 Report
Sanoguera-Miralles and Bueno-Martinez et al. present an interesting manuscript that utilizes a minigene splicing assay to examine twenty RAD51C variants catalogued through ClinVar. All of the variants examined are found within 4 bp of the canonical splice site. Using the minigene assay the authors were able to demonstrate that all 20 variants altered splicing of the minigene compared to wildtype constructs. While some of these variants resulted in altered transcripts that likely disrupt protein function through either nonsense mediated RNA decay or protein truncation, the authors demonstrate in-frame exon skipping in others. Leaking splicing was also noted in two instances resulting in detection of the canonical transcript in 26.3% and 21.3% of transcripts, respectively.
This was a very interesting manuscript that was well written. The minigene splicing assay is simplistic in its design, yet provides important functional evidence for the evaluation of spliceogenic variants. While in silico prediction tools can provide insights to possible impacts of nucleotide variants on splicing, these prediction tools do not replace the empirical evidence that the minigene assay can provide. The type of work presented in this manuscript is invaluable for accurate interpretation of spliceogenic variants that may be detected in families with suspected hereditary breast/ovarian cancer. The assay used is well established and has been used previously to evaluate spliceogenic variants in RAD51C and other hereditary breast/ovarian cancer-associated genes. The transcript analysis results demonstrated splice complexity with several variants resulting in two or more transcript isoforms. The assay results enabled direct measurement of the percentage of the isoforms generated thus facilitating a better understanding of the dominant isoform generated by each variant and their effect on protein coding.
The authors further used the minigene splicing assay results, and understanding of RAD51C protein biology, to evaluate each variant using the most current ACMG-AMP frameworks for variant classification. These variant interpretations were generally evidence based and well thought out.
In general, I thought that this manuscript was interesting to read and added important information to the literature to facilitate accurate variant interpretation of RAD51C spliceogenic variants, which in turn can better facilitate risk assessment and genetic counselling in families carrying these variants. I only had minor suggestions or comments:
Line 156 – the authors refer to “ad-hoc rules” that they have developed for consideration of the different coding transcripts associated with the same spliceogenic variant in variant interpretation and classification. Although a reference is provided so that the reader can look up what these ad-hoc rules are, it would also be helpful to briefly describe these in the current manuscript.
Lines 250-251 and 263-265– In these lines the authors discuss transcripts in which the encoded proteins lack some beta strands. The lack of these protein structures are used as evidence to support pathogenicity. It is not clear from the text how lack of these beta strands is predicted to impact protein function. Is there evidence from another source that these beta strands are critical to protein function and that their loss is deleterious (rather than resulting in normal or slightly reduced protein activity)?
Lines 259-274 – This paragraph refers to evidence used in the variant interpretation of three aberrant transcripts that kept the open reading-frame (Δ(E2p3), Δ(E5) and ▼(E8p3)). However, the corresponding tables that summarizes variant classification according to the ACMG/AMP-based criteria (Table 2), does not include these transcript isoform names. As such, in order to correlate the discussion in this paragraph with the information in table 2, the reader also needs to cross reference Table 1 or 3. Incorporation of the transcript isoform names in table 2 would assist the reader in correlating this discussion of transcript isoforms with the corresponding evidence used to classify each of the variants.
Lines 342-359 – This paragraph discusses two RAD51C variants, c.404+3A>G and c.705+3A>G, for which mg-FL transcripts were detected in 26.3% and 21.3% or transcripts, respectively. The discussion of these variants in this section (and in other sections) does not address the significance of canonical transcripts in these cases. Leaky splice variants have been reported in various genes and sometimes can be associated with milder phenotypes (or no phenotypes), presumably because the canonical isoforms contribute to a “phenotypic rescue”. Is it known whether there is a threshold of RAD51C deficiency that is tolerated before associated cancer risks become increased?
Thank you for the opportunity to review this interesting manuscript.
Author Response
- Thank you very much for the positive comments and the revision of our manuscript.
1 - Line 156 – the authors refer to “ad-hoc rules” that they have developed for consideration of the different coding transcripts associated with the same spliceogenic variant in variant interpretation and classification. Although a reference is provided so that the reader can look up what these ad-hoc rules are, it would also be helpful to briefly describe these in the current manuscript.
- Acknowledge this comment. We have introduced new several sentences in this paragraph of Materials and Methods, which define these ad-hoc rules.
Lines 154-155:
“mgR51C read-outs have been introduced into the classification system as PVS1_O or BP7_O codes of variable evidence strength depending on the splicing outcome [P, Sup-porting (±1 point); M, Moderate (±2); Strong (±4); Very Strong (±8)].”
Lines 163-179:
“we have developed some ad-hoc rules that take into consideration the coding potential of each individual transcript and its relative contribution to the overall expression to reach the appropriate PVS1_O or BP/_O evidence strength. In brief, for each complex read-out we have applied the following algorithm: (i) De-convolute mgR51C read-outs in-to individual transcripts; (ii) apply ACMG/AMP evidences to each individual transcript; (iii) produce an overall PVS1_O (or BP7_O) code strength based on the relative contribu-tion of individual transcripts/evidences to the overall expression. Thus, if pathogenic supporting transcripts contribute ≥90% to the overall expression, PVS1_O_ code is applied (if different transcripts support different pathogenic evidence strengths, the lowest strength contributing >10% to the overall expression is selected for overall evidence strength). Similarly, BP7_O_ code is applied if benign supporting transcripts contribute ≥90% to the overall expression (if different transcripts support different pathogenic evi-dence strengths, the lowest strength contributing >10% to the overall expression is selected for overall evidence strength). If neither pathogenic nor benign supporting transcripts contribute ≥90% to the overall expression, the splicing assay is considered not providing any evidence in favor, or against, pathogenicity. Recently, we have used a similar ap-proach to deal with complex PALB2/ATM minigene read-outs [20,30].”
2 - Lines 250-251 and 263-265– In these lines the authors discuss transcripts in which the encoded proteins lack some beta strands. The lack of these protein structures are used as evidence to support pathogenicity. It is not clear from the text how lack of these beta strands is predicted to impact protein function. Is there evidence from another source that these beta strands are critical to protein function and that their loss is deleterious (rather than resulting in normal or slightly reduced protein activity)?
- Acknowledge this comment. The five RAD51 paralogs are known to be required for homologous recombination and maintenance of genomic stability. Indeed, RAD51C interacts with RAD51B, RAD51D, XRCC2 and XRCC3 in two different complexes that play a role in homologous recombination. Miller et al studied the interaction between RAD51B and D (and also XRCC3) with deletion mutants. These authors found that Rad51C1-285 (includes β-strands 1-5) or Rad51C285-376 (includes β-strands 6-9) did not bind RAD51B. So, a complete beta-sheet is important in maintaining the overall fold of the protein. Moreover, the missense variant p.Arg312Trp (ß-strand 6) has been shown to impair RAD51C function (Gayarre et al 2017).
Both studies indicate that this protein region is essential for RAD51C function so that transcripts lacking any of the β-strands, such as ▼(E6q4)-a, ▼(E6q4)-b, Δ(E7), Δ(E8) or the in-frame isoform Δ(E5), is probably deleterious.
- We have modified this part, adding several sentences to clarify it.
Lines 270-277:
“ The integrity of the β-sheet is important in maintaining the overall fold of the RAD51C protein and the interaction with RAD51B, so that alterations of any ß-strand of RAD51C should be considered deleterious [33]. Further, structural features (the order of the ß--strands in space is not the same as their order in sequence) predict that proteins lacking any single b-strand would fail to form the ß--sheet resulting in a collapse of the protein core and misfolding of the protein [33]. Moreover, the missense variant p.Arg312Trp (ß-strand 6) has been shown to impair RAD51C function [34]. Altogether these data,…”
3- Lines 259-274 – This paragraph refers to evidence used in the variant interpretation of three aberrant transcripts that kept the open reading-frame (Δ(E2p3), Δ(E5) and ▼(E8p3)). However, the corresponding tables that summarizes variant classification according to the ACMG/AMP-based criteria (Table 2), does not include these transcript isoform names. As such, in order to correlate the discussion in this paragraph with the information in table 2, the reader also needs to cross reference Table 1 or 3. Incorporation of the transcript isoform names in table 2 would assist the reader in correlating this discussion of transcript isoforms with the corresponding evidence used to classify each of the variants.
- We have added all the transcript names and their contribution in Table 2 (Column PVS1_O/BP7_O mgR51C_ex2-8).
- We have included cross-references to Table 1 to facilitate understanding of the manuscript.
4- Lines 342-359 – This paragraph discusses two RAD51C variants, c.404+3A>G and c.705+3A>G, for which mg-FL transcripts were detected in 26.3% and 21.3% or transcripts, respectively. The discussion of these variants in this section (and in other sections) does not address the significance of canonical transcripts in these cases. Leaky splice variants have been reported in various genes and sometimes can be associated with milder phenotypes (or no phenotypes), presumably because the canonical isoforms contribute to a “phenotypic rescue”. Is it known whether there is a threshold of RAD51C deficiency that is tolerated before associated cancer risks become increased?
-Very important comment. As the reviewer indicates, it would be essential to define the threshold of RAD51C expression from which it keeps its tumor suppressor activity. Unfortunately, it is not known by now but this finding would provide critical information to determine the pathogenicity of leaky spliceogenic variants.
- So, we have modified this paragraph to introduce this information:
“Only two variants (c.404+3A>G and c.705+3A>G) displayed 26% and 21% of the mgFL-transcript, respectively. Unfortunately, it is not known the minimal amount of RAD51C expression to confer tumor suppressor haplosufficiency so, these splicing assays were not considered informative (PVS1_O_N/A).”
Note that leaky variants generate complex minigene read-outs (two or more transcripts), and are therefore classified accordingly (see methods).
Reviewer 2 Report
The manuscript entitled "Minigene splicing assays identify 20 spliceogenic variants of the breast/ovarian cancer susceptibility gene RAD51C authored by Sanoguera-Miralles et al., is a very interesting study. It reports novel results and deserves publication in Cancers. Actually, not many publications report variants affecting splicing of RAD51C. The manuscript is very well written, clear and well discussed; I have just few comments: in the Introduction in the sentence 61-69 the authors do not mention anything about BRCA1 (they mention MLH1 though); I would expect that as BRCA1 splice variants are deeply studied. In the Materials and Methods, the section 2.6 is not actually a section that described some methodology. It appears to be more appropriate as results and maybe some information could be in the introduction. The acronyms need to be explained. Not all readers are familiar with PTC (premature termination codon) and NMD (nonsense mediated decay); as some variants are found PTC-NMD, this needs to be explained better. I would also stress a little more that FL is almost undetectable.
In conclusion, the manuscript is acceptable after minor revision
Author Response
Thank you very much for the positive comments
1- in the Introduction in the sentence 61-69 the authors do not mention anything about BRCA1 (they mention MLH1 though); I would expect that as BRCA1 splice variants are deeply studied.
- Acknowledge this suggestion. We have added two references of BRCA1 and BRCA2 studies.
2- In the Materials and Methods, the section 2.6 is not actually a section that described some methodology. It appears to be more appropriate as results and maybe some information could be in the introduction.
- Acknowledge this comment. In this section we have tried to describe the classification approach and the rules we have followed to classify the variants. Certainly, we agree with the referee as sometimes the method is mixed with some results. So, we have moved some sentences of the last paragraph of Materials and Methods to Results.
“The PM3 evidence (in trans with a pathogenic variant in a recessive disorder) did not con-tribute to the final classification. Not surprisingly (FANCO is an extremely rare FA complementation group) [37], none of the tested variants has been identified in Fanconi Anemia patients (ClinVar and Global Variome shared LOVD databases and literature search-es). Similarly, the BS2 evidence (in trans with a pathogenic variant in a healthy individual) does not contribute to the final classification of our tested variants. Finally, we have considered that some pathogenic (PS2, PM1, PM6, PP2, PP4) and benign (BP1, BP3, BP5) codes are not applicable to the classification of RAD51C variants.”
3-The acronyms need to be explained. Not all readers are familiar with PTC (premature termination codon) and NMD (nonsense mediated decay); as some variants are found PTC-NMD, this needs to be explained better. I would also stress a little more that FL is almost undetectable.
-We have included explanations of these acronyms in text and Table 1 and have modified the following sentence:
“Of the 19 characterized transcripts, 14 introduced premature termination codons (PTC; PTC transcripts), and of these, 10 are predicted to be degraded by the Nonsense-Mediated Decay pathway (NMD; PTC-NMD transcripts) that is considered convincing evidence of deleteriousness (Supplementary Table S3). “
- We have modified the sentence of the Fl-transcript:
“All variants altered splicing, 18 of which produced no traces of the mgFL-transcript or almost undetectable levels (<2.4%, c.904G>A)…”
Reviewer 3 Report
The manuscript "Minigene splicing assays identify 20 spliceogenic variants of the breast/ovarian cancer susceptibility gene RAD51C" by Sanoguera-Miralles et al. describes about the analysis of RAD51C variants using mini-gene assays for splicing defect. Many nucleotide variants, whether they affect protein changes or not can affect potential splicing of exons. In the analysis of variants it is very relevant. The authors used bioinformatic analysis to sort out variants that potentially affect splicing and then analyzed them using mini-gene assays. Although mini-gene assays sometimes is not reflective for the splicing of whole gene in a particuar locus but it is now is an well accepted preliminary screening assay. It is difficult to call the variants as deleterious or neutral based on only splicing assays. Overall manuscript is well written, very relevant for variant analysis and is suitable for publication if they address the following concerns:
1) Splicing variants are called solely based on the size difference. It is good to look into Sanger sequence of splice variants after gel elution of the band and sequencing.
Author Response
Thank you very much for the positive comments
1) Splicing variants are called solely based on the size difference. It is good to look into Sanger sequence of splice variants after gel elution of the band and sequencing.
Acknowledge this comment.
We have sequenced the RT-PCR products of all variant assays (indicated in Materials and Methods, section 2.5. Minigene Splicing Assays). In fact, all the *.ab1 sequence and*.fsa fragment analysis files of RT-PCR products will be freely available at http://hdl.handle.net/10261/270934; https://doi.org/10.20350/digitalCSIC/14662 upon manuscript acceptance (links indicated in the manuscript section “Data Availability Statement”).
Unfortunately, Sanger sequencing only allowed us to characterize the main transcripts, while the minor ones (<10% of the overall expression) are really difficult to characterize since gel band extraction does not work properly with these small amounts, or other methods (e.g. subcloning of RT-PCR products into a PCR-vector, or RNAseq of minigene outcomes) are laborious and not cost-effective.
Anyway, we have also been using Fluorescent fragment analysis for transcript characterization in our previous studies (since Acedo et al, 2012). We have shown that this technique is highly sensitive, accurate and shows high resolution. For example, in Figure 1c, transcripts with minimal size differences (1-3 nt) are well-discriminated. So, for minor-rare transcripts is a good option (not perfect, we agree with the reviewer) to annotate them.
- To clarify it, we have modified this part:
Lines 132-134:
“RT-PCR products were sequenced by Macrogen (Madrid, Spain), which allowed the characterization of the main variant-induced transcripts. Minor transcripts were annotated according to fluorescent fragment electrophoresis size data (see below).”